# Induction and Cancellation of Self-Motion Misperception by Asymmetric Rotation in the Light

**Vito Enrico Pettorossi [1,\*], Chiara Occhigrossi [1], Roberto Panichi [1] , Fabio Massimo Botti [1], Aldo Ferraresi [1], Giampietro Ricci [2] and Mario Faralli [2]**

[1] Department of Medicine and Surgery, Section of Human Physiology, University of Perugia, 06132 Perugia, Italy
[2] Department of Medicine and Surgery, Section of Otorhinolaryngology, University of Perugia, 06132 Perugia, Italy
[\*] Correspondence: vito.pettorossi@unipg.it

**Abstract:** Asymmetrical sinusoidal whole-body rotation sequences with half-cycles at different velocities induce self-motion misperception. This is due to an adaptive process of the vestibular system that progressively reduces the perception of slow motion and increases that of fast motion. It was found that perceptual responses were conditioned by four previous cycles of asymmetric rotation in the dark, as the perception of self-motion during slow and fast rotations remained altered for several minutes. Surprisingly, this conditioned misperception remained even when asymmetric stimulation was performed in the light, a state in which vision completely cancels out the perceptual error. This suggests that vision is unable to cancel the misadaptation in the vestibular system but corrects it downstream in the central perceptual processing. Interestingly, the internal vestibular perceptual misperception can be cancelled by a sequence of asymmetric rotations with fast/slow half-cycles in a direction opposite to that of the conditioning asymmetric rotations.

**Keywords:** self-motion perception; contrast velocity stimulation; perceptual adaptation; vestibular misperception; perceptual vestibular recovery

## 1. Introduction

It has recently been shown that an adaptive effect in self-motion perception is induced when subjects undergo a repetitive horizontal sinusoidal rotation in which the contrast velocity stimulus is delivered in the dark with a sequence of fast and slow half-cycles [1–4]. The adaptation consists of a gradual change in self-motion perception that decreases during slow rotation and increases during fast rotation, enhancing the asymmetry of the motion perception and causing an incorrect estimation of body position. This effect persists over time, conditioning the subsequent perceptual responses. In contrast, the vestibulo-ocular reflex is differently affected by asymmetric rotation, because the gain of the response to slow rotation increases and that to fast rotation decreases [2], reducing the asymmetry of the reflex. The dissociation between perception and reflex may be the result of different elaboration of vestibular signals involving separate central pathways [5–17].

In physiological conditioning, enhanced adaptive asymmetry of self-motion perception can occur during repeated movements at different velocities or paths along a curved trajectory or rotations in sport activities, and induces an expansion of the dynamic resolution of the system, which may be adequate to better detect fast body rotation and extract information relevant to the "impending" straight-ahead or body position. Conversely, the adaptive central process of the vestibulo-ocular reflex tends to reduce the circuital asymmetry for diminishing the side imbalance and improving the gaze stability [2]. The decreased sensitivity to slow movements does not necessarily represent a functional deficiency, because other sensory modalities such as vision and proprioception can compensate for the reduction in vestibular responses [1]. The fact that the vestibular system tends to

perceive fast movements better than slow movements was also evidenced by a study in which fast and slow oscillations were delivered simultaneously [15].

The adaptive motion perceptual changes, however, in the presence of functional alterations of the vestibular system, as in unilateral labyrinth deficits [18,19], can become maladaptive. Indeed, following unilateral vestibular deficit, it has been shown that self-motion misperception can persist over time, even when the vestibulo-ocular reflex has fully recovered [18,19]. It has been suggested that the unilateral deficit determines an asymmetric vestibular input that lasts throughout the chronic post-lesion period, during which the dynamic vestibular input from one side is reduced. In fact, the vestibular input is greater when rotation is toward the healthy side than when it is toward the lesion side. This may explain the persistence of discomfort in vestibular patients, even when the vestibulo-ocular reflex is fully recovered. However, the induction and maintenance of a maladaptive mechanism implies that vision is not able to abolish the effect of the erroneous motion perception of vestibular origin. Therefore, the explanation for the perceptual disturbance of patients that persists over time [18] must assume that broader visual input is unable to suppress the vestibular system's adaptive mechanism.

The aim of this study was to examine whether visual input is effective in overcoming adaptive vestibular perception using a conditioning paradigm. The perceptual adaptation was tested after conditioning with asymmetric vestibular stimulation in the dark and in the presence of visual input. A further aim of the study was to demonstrate whether the erroneous vestibular perception, once induced, could be cancelled by oppositely directed asymmetric vestibular stimulation.

## 2. Material and Methods

### 2.1. Participants

Twelve healthy subjects aged 25–45 years (7 men and 5 women) participated in the study after providing written informed consent. The experimental protocol was in accordance with the Declaration of Helsinki (1964) and was approved by the local ethical committee. The subjects included in the study reported no vestibular symptoms in the past and showed normal responses to conventional vestibular tests for horizontal vestibular canals and normal orientation of subjective visual vertical. They were also able to perform the motion perception test in response to asymmetrical stimulation without difficulty, showing a perceptual vestibular error in the normal range [2], indicating that they were able to perform the perceptual tests correctly.

### 2.2. Test of Self-Motion Perception: Stimulation Apparatus and Recording

2.2.1. Stimulation Apparatus

Subjects sat in the center of a computer-controlled rotating chair in an acoustically isolated cabin with a 100 cm radius (Figure 1). The horizontal rotation of the chair was driven by a DC motor (Powertron, Contraves, Charlotte, NC, USA) servo-controlled by an angular-velocity encoder (0.01–1 Hz, 1% accuracy). A holder maintained the head tilted down at 30° and aligned with the rotation axis of the platform. The trunk was tightly fastened to the chair. Roll and pitch head displacement was prevented by a plastic collar.

2.2.2. Self-Motion Perception Conditioning

A conditioning procedure was used to induce fast/slow perceptual adaptation. Subjects underwent four cycles of asymmetric horizontal whole-body oscillations in the dark or in the light. The chair was rotated back and forth at different velocities [2] to activate mainly the subject's horizontal semicircular canals. The stimulus profile of the chair movement resulted from the combination of two sinusoidal half-cycles of the same amplitude (40°), but different frequencies: fast half-cycle (FHC) = 0.40 Hz and slow half-cycle (SHC) = 0.09 Hz (Figure 2). Peak acceleration during the fast hemicycle was $120°/s^2$ with a peak velocity of 47°/s, followed by slow rotation in the opposite direction at a peak acceleration of $7°/s^2$, with a peak velocity of 11°/s, which returned the subject to the starting posi-

tion. Both acceleration and velocity values are well above the thresholds for vestibular activation [20–22].

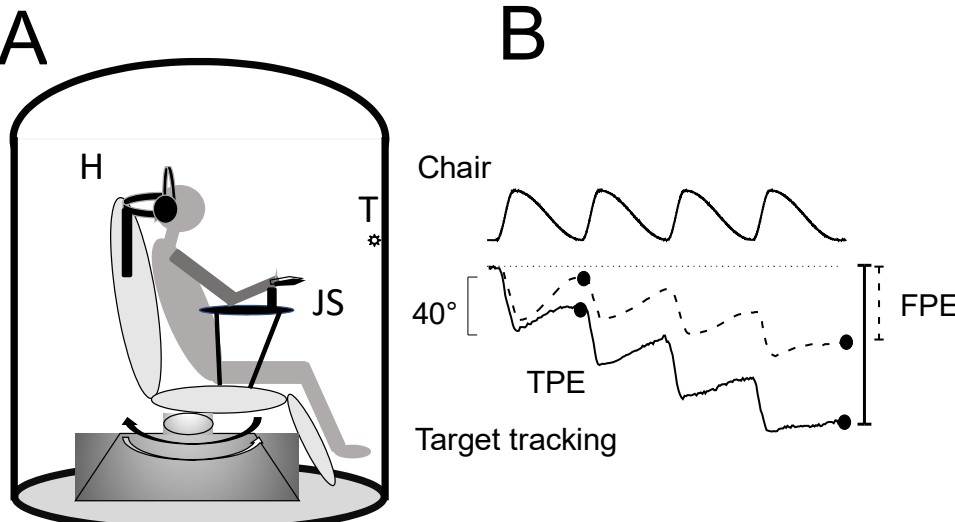

**Figure 1.** (**A**) Schematic drawing of the experimental setup. H: holder that keeps the head, trunk, and pelvis in place during cyclically imposed whole-body yaw rotations produced by rotation of the chair; JS: joystick for tracking position recording; T: visual target. Solid and outlined arrows indicate fast and slow rotation, respectively. (**B**) Tracking of perceived body motion in response to asymmetric chair rotation before (dashed line) and after (entire line) conditioning. Black spots indicate tracking position error (TPE) and final position error (FPE).

### 2.2.3. Self-Motion Perception Testing

Four cycles of asymmetric whole-body oscillations were performed in the dark to test the effect of previous conditioning asymmetric stimulation or cancellation. The profile of the testing rotation was the same of that of the conditioning stimulation.

Self-motion perception recordings. We used a psychophysical tracking procedure to assess self-motion perception [23,24]. Before starting the rotation, subjects stared at the target placed in front of them and were asked to continue to imagine the same target throughout the rotation in the dark with eyes closed. The target was a spot of light (diameter 1 cm) projected onto the wall of the dark cabin 1.5 m from the subject's eyes. The light was switched off just before the onset of rotation and switched back on at the end of rotation. Subjects were instructed to continuously track the remembered spot in the dark by counter-rotating the hand pointer connected to a precision potentiometer (joystick) (Figure 1).

During asymmetric rotation, subjects perceived the fast hemicycle more vividly than the slow hemicycle [2], so, at the end of each cycle, the target was erroneously represented as being in the direction of the slow hemicycle (tracking position error, TPE). The final tracking position error (FPE) after four cycles resulted from the algebraic sum of single cycle errors plus additional adaptation that further enhanced the perception of fast rotation and reduced that of slow rotation [2]. The amplitude of the position error of the single cycle (TPE) and the final position error (FPE) at the end of four cycles was evaluated.

### 2.2.4. Protocol for Conditioning and Testing Procedure

The subjects underwent a training session with the tracking system used in the self-motion perception experiment. In this process, subjects were trained to track first the light spot and then the remembered spot with the pointer during asymmetric rotations. For quality control of the manual tracking, the pointer trace was examined. We stopped the training when visual inspection of the tracking showed no further improvement and there was

good matching of the stimulus waveform. To avoid any carryover effect, the conditioning and testing sessions were performed on separate days. After training, subjects underwent different stimulation procedures in four separate sessions administered randomly.

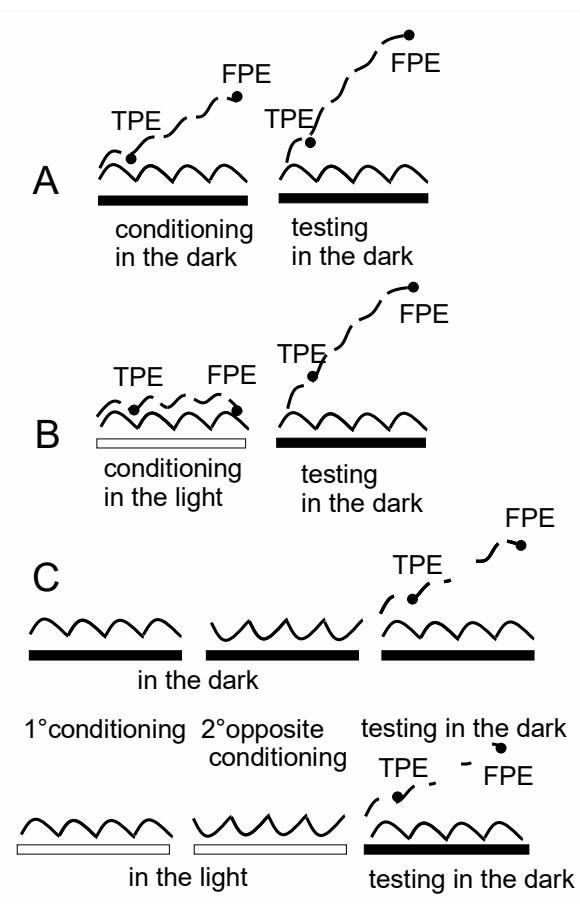

**Figure 2.** Stimulation procedure for conditioning and cancellation. (**A**) Four asymmetric cycles in the dark for conditioning followed by four asymmetric cycles in the dark for testing session 1; (**B**) four asymmetric cycles in the light for conditioning followed by four asymmetric cycles in the dark for testing; (**C**) four asymmetric cycles for conditioning plus opposite asymmetric cycles for cancellation and four asymmetric cycles for testing (same conditioning asymmetry). Solid line: asymmetric rotation; dashed line: perceptual tracking. Horizontal bar: white: light on; black: light off. Black spot on tracking trace after the first cycle indicates TPE and that at the end of the four cycles indicates FPE.

Stimulation for conditioning was carried out in the light and dark, while test stimulation was only carried out in the dark to reveal the effect of conditioning. Otherwise, the visual input would have cancelled out the perceptual error completely. The sessions were as follows (Figure 2): session 1: four cycles of conditioning asymmetric rotation in the dark and, after 3 min, four cycles of testing asymmetric rotation in the dark to confirm that asymmetric rotation causes persistent perceptual adaptation; session 2: four cycles of conditioning asymmetric rotation in the light and, after 3 min, four cycles of testing asymmetric rotation in the dark to show whether or not light prevents the conditioned adaptation to asymmetric stimulus; session 3: four cycles of conditioning asymmetric rotation followed by four cycles of fast and slow asymmetric rotation in the opposite direction to the previous rotation, with all rotations in the dark, and then four cycles of testing asymmetric rotation in the dark—this session is for evidencing whether the conditioned adaptation is cancelled by opposite rotations in the dark; session 4: four cycles of conditioning asymmetric rotation followed by four cycles of fast and slow asymmetric rotation in the opposite direction to the

previous rotation, with all rotations in the light, and then four cycles of testing asymmetric rotation in the dark—this session is for evidencing whether the conditioned adaptation is cancelled by opposite rotations in the light. The subjects performed the tracking during the conditioning and testing procedures.

The persistence of the conditioned effects was also examined by testing the perception of asymmetric rotation after increasing the interval of the conditioning procedure (5, 10, 15, 20, 25, 30 min).

*2.3. Data Evaluation and Statistical Analysis*

The amplitudes (mean and SD) of TPE and FPE are reported. Statistical analysis was performed using repeated-measures ANOVA. The power values for all analyses are reported as η. Exponential functions were used to fit the values obtained after the cycles during the sequence of rotation. R and $X^2$ values indicate the goodness of the exponential fit.

The level of significance was set at $p < 0.05$. Prior to ANOVA, the *W* test [25] was used to assess the normality and Levene's test was used to assess the homogeneity of variances. All statistical evaluations were performed with OriginPro software (Origin Lab Corporation, Northampton, MA, USA).

## 3. Results

*3.1. Misperception of Self-Movement Induced by Four Cycles of Asymmetric Rotation in the Dark*

Self-motion perception in the dark was examined by measuring the amplitude of tracking during the four asymmetric rotations by 12 individuals (session 1). The amplitude of tracking during the FHC of the first rotation was $34 \pm 4°$, and that of the subsequent SHC was $29 \pm 6°$ (Figure 3).

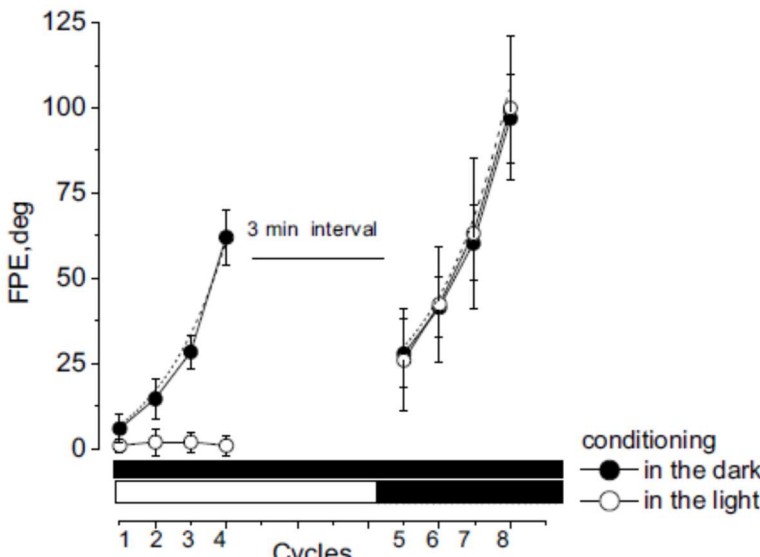

**Figure 3.** Tracking of the remembered target during cycles of asymmetric rotation (session 1 and 2). Left graph: four cycles of conditioning; right graph: four cycles of testing. Amplitude of final position error with conditioning in the dark (filled circles) and in the light (open circles). Data are reported as mean and SD. Dashed line represents the exponential fit. Bars below indicate the lighting condition during conditioning and testing: black indicates dark and white indicates light.

As a result, at the end of the first cycle, the body position was misperceived, showing a shift of $5 \pm 4°$ from the center toward the side of the slow rotation (tracking position error, TPE). The amplitude of FHC responses increased in successive asymmetric cycles, while the amplitude of SHC decreased to almost zero (Figures 3 and 4). The TPE of each cycle increased exponentially so that the final position error (FPE) at the end of the four cycles was

64 ± 8° (t = 6.56 s, R = 0.99, $X^2$ = 1.48). This large error is the consequence of asymmetrical conditioning, which induces adaptive responses by increasing and decreasing responses to fast and slow rotation, respectively. The amplitude of the TPE and FPE observed after repetitive asymmetric rotation was consistent with that observed in previous studies (1–3).

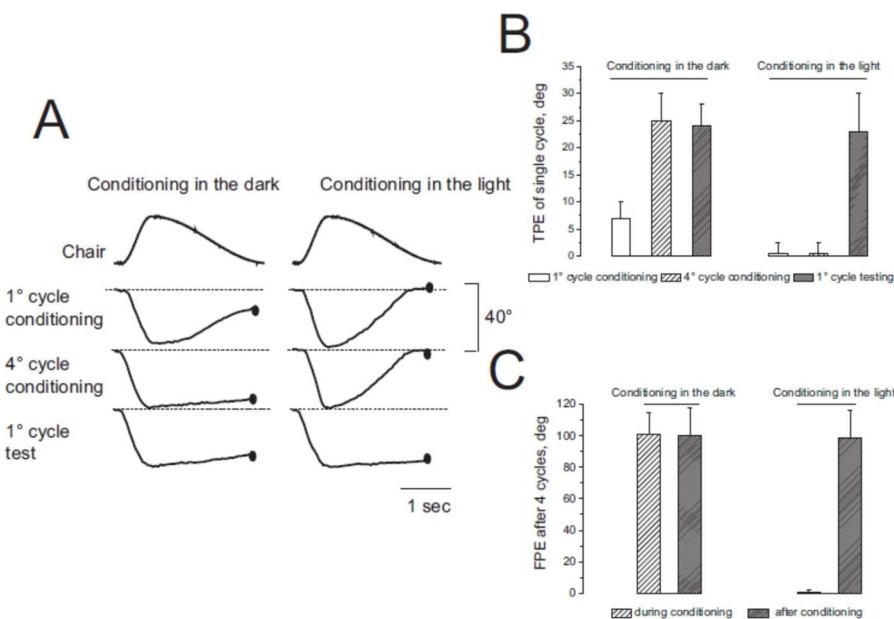

**Figure 4.** Tracking of the visual and remembered target. (**A**) Left and right columns show the single cycle response to conditioning in the dark and, respectively (session 1 and 2) (the lines indicate the experimental light condition). The distance between tracking at the end of rotation (black spot) and position of tracking before rotation is the tracking position error (TPE). (**B**) TPE of the response to a single cycle (1°, 4° conditioning cycle, and 1° test cycle). (**C**) FPE of the response to four cycles. In (**B**,**C**), data in the left graph are from conditioning in the dark and those in the right graph are from conditioning in the light. Columns represent the mean and SD of errors of 12 individuals.

After a 3 min interval, four cycles of asymmetric rotations were repeated in the dark to test the perceptual effect of the previous four cycles of asymmetric rotations (conditional procedure). The test procedure showed that the TPE of the responses to the first single asymmetric cycle (25 ± 12°) was significantly higher compared with the first cycle of the conditioning procedure (F $_{(1-22)}$ = 24.75, $p$ < 0.001, η = 0.98) (Figures 3 and 4). This error increased exponentially in subsequent cycles (t = 11.22 s, R = 0.98, $X^2$ = 1.11) so that the FPE after four cycles increased to 104 ± 12° (F $_{(1-22)}$ = 81.22, $p$ < 0.001, η = 0.96), significantly greater than that observed at the end of conditioning. The FPE remained higher for at least 30 min after conditioning (Figure 5).

*3.2. Misperception of Self-Movement Induced by Four Cycles of Asymmetrical Rotation in the Light*

Conditioning with four cycles of asymmetrical rotation was repeated in the light for all individuals (session 2). The perception of self-movement during rotation was almost correct and the amplitude of motion perception corresponded to the amplitude of the stimulus, which was similar to that of the rotation chair (40 ± 2° during FHC and 40 ± 3° during SHC). The TPE of each cycle was always zero (0 ± 3°) and, consequently, the FPE at the end of the four cycles was also zero (0.5 ± 3°) (Figures 3 and 4). After a 3 min interval, the four cycles of asymmetric rotation were repeated in the dark to test the perceptual effect of the previous conditioning in the light.

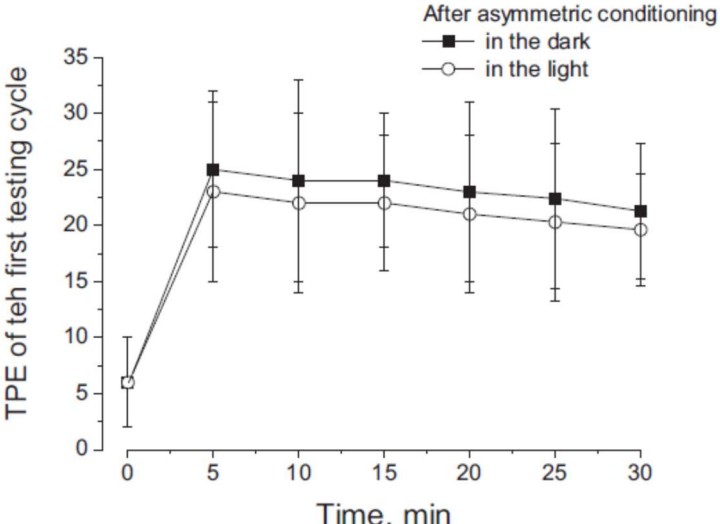

**Figure 5.** Time course TPE in response to the first cycle of testing asymmetric rotation delivered at different intervals from the conditioning procedure (time = 0). Data are the mean and SD of TPE.

Perceptual responses to each asymmetric cycle showed TPE due to a reduction in perceptual responses to SHC and an increase during FHC ($F_{(1-22)}$ = 104.84, $p < 0.001$, $\eta$ = 0.99). The TPE of the first testing cycle was 24.5 ± 7°, statistically different from the first cycle of conditioning, when the value was close to 0° ($F_{(1-22)}$ = 104.84, $p < 0.001$, $\eta$ = 0.99) (Figures 3 and 4). The TPE increased exponentially in response to subsequent cycles (t = 10.2 s, R = 0.99, $X^2$ = 1.53), so that the FPE after four cycles increased to 97 ± 20°, significantly larger than that observed at the end of the conditioning procedure ($F_{(1-22)}$ = 308, $p < 0.001$, $\eta$ = 0.98). In addition, the statistical comparison between the TPE of the first testing cycle observed in the dark and in the light showed no statistical significance ($F_{(1-22)}$ = 0.90, $p > 0.21$, $\eta$ = 0.11). Similarly, the comparison between FPE in the tests after conditioning in the light and the dark showed no statistically significant difference ($F_{(1-22)}$ = 0.98, $p > 0.33$, $\eta$ = 0.17). The conclusion is that, although the previous conditioning stimulation did not induce any position error when the asymmetric rotation was performed in the light, the subsequent asymmetric testing rotation induced the same large TPE and FPE as those observed when the conditioning was performed in the dark.

The perceptual error after light conditioning was maintained for at least 30 min, as in the dark conditioning (Figure 5).

*3.3. Cancellation of Perceptual Mismatch by Opposite Directed Asymmetric Rotation in the Dark and Light*

Immediately after the asymmetric conditioning rotation, four cycles of directly opposite asymmetric rotation were administered in the dark, with the direction of fast and slow half-cycles opposite to that of the asymmetric conditioning rotation (session 3). After 3 min, the effect of this procedure was tested by four cycles of asymmetric rotation. The values of TPE and FPE were significantly different from those observed in the conditioning rotation: the TPE amplitude of the first test cycle was 9.3 ± 4.2° ($F_{(1-22)}$ = 0.88, $p > 0.24$, $\eta$ = 0.19) and the FPE amplitude was 68.4 ± 11.1° ($F_{(1-22)}$ = 0.92, $p > 0.37$, $\eta$ = 0.13) (Figure 6).

These results indicate that opposite asymmetrical rotation administered after conditioning cancelled the effect of the previous conditioning procedure. This cancellation was also observed when conditioning and directly opposite asymmetric rotation were administered in the light (Figure 5) (session 4): the TPE of the first cycle was 8.5 ± 4.3° ($F_{(1-22)}$ = 0.98, $p > 0.45$, $\eta$ = 0.11) and the FPE was 61 ± 9° ($F_{(1-22)}$ = 0.87, $p > 0.31$, $\eta$ = 0.18), which are not different from the conditioning value.

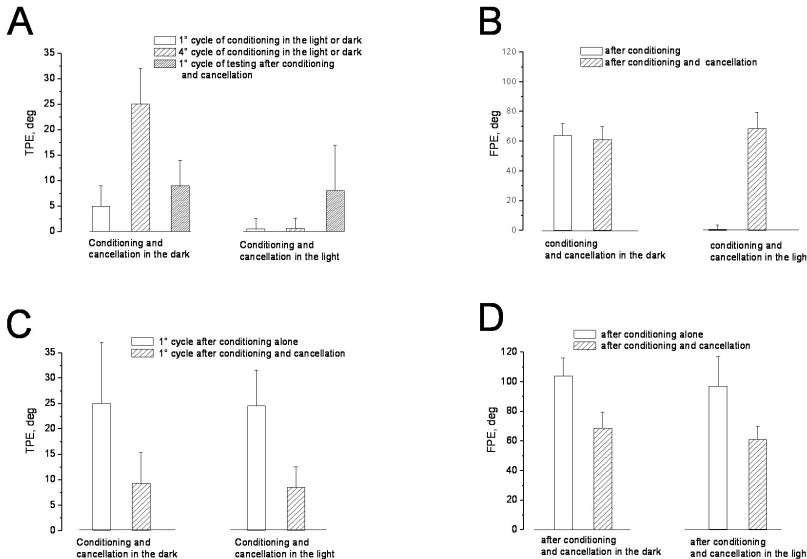

**Figure 6.** TPE and FPE induced by conditioning and conditioning plus cancellation procedures in dark and light conditions (session 3 and 4). Column values represent the mean and SD of perceptual error. (**A**) TPE of 1° and 4° conditioning cycles (two left columns) and 1° testing cycle after conditioning and cancellation (right column). (**B**) FPE at the end of the conditioning procedure (right column) and the end of testing after conditioning plus cancellation (left column). (**C**) Comparison between TPE of one testing cycle after conditioning (left column) and after conditioning plus cancellation in the dark and light (right column). (**D**) Comparison of testing FPE after conditioning alone (left column) and after conditioning plus cancellation (right column).

## 4. Discussion

The first result of this study is that sinusoidal rotation of the whole body in the horizontal plane at different half-cycle velocities in the dark causes perceptual adaptation in the vestibular system that leads to the absence and enhancement of slow and fast motion perception, respectively [2]. This confirms previous observations of the response to asymmetric rotation performed in the dark but using a different protocol [1–4]. This effect may be the result of intermittent activation of a velocity storage mechanism in the perceptual circuitry of the horizontal semicircular canals, a mechanism like that observed in the oculomotor system. However, the perceptual effect of the intermittent activation is much more durable. The perception change persists for more than 30 minutes and displaces the internal representation of the body position in space, while the velocity storage observed in the ocular responses is remarkable shorter. This highlights that the central elaboration of the vestibular signals directed to the motion perception is different from that directed to the oculomotor system.

In addition, an important new result is reported here: the effect was also induced when conditioning asymmetric rotation was performed in the light. The fact that the adaptive vestibular misperception showed the same magnitude as that observed after conditioning in the dark is surprising, because, in the presence of broader visual input, the vestibular perceptual error in response to asymmetric rotation was not observed because of perfect perceptual pursuit during fast and slow half-cycles. In fact, the body position relative to the visual target was correctly represented at the end of the conditioning rotations. The effect of conditioning in the dark and light remained for at least 30 min. Apparently, there is no misperception of body motion in light, but the motion perception error remains hidden. It is likely that vision is not able to override the adaptive vestibular process but prevents it downstream in the self-motion perceptual process. The hidden vestibular mismatch could interfere with behavior, because the visual and vestibular signal are not consistent. There is no direct evidence for the effects of this visual–vestibular discrepancy, but it can be hypothesized that, given the importance of the vestibular system in orientation, complex

motor functions, and cognition [9,26–29], a visual–vestibular mismatch may induce various functional alterations.

The higher sensitivity of the vestibular system to fast movements in combination with the visual and proprioceptive systems, which are sensitive to slow movements [1,30,31], is functionally useful to enhance the self-motion perception. However, this functional advantage can turn into a disadvantage when the asymmetric stimulus is prolonged, as in the case of unilateral vestibular deficit. In this case, rotation to the healthy side will provide much more intense signals than rotation to the injured side, introducing an asymmetry in the input that is maintained. This can produce a consolidation of the adaptive mechanism that, in this case, turns out to be an incorrect adaptation. This may be relevant, because perceptual errors of vestibular origin may occur with several unilateral vestibular deficits, and often they may be responsible for persistent patient discomfort [18,19]. In these studies, we showed that self-motion perception impairment persists for a long time in patients with vestibular neuritis, sometimes even longer than the reflex impairment, even though vision can mask the internal vestibular perceptual error. Indeed, when there is a chronic unilateral vestibular system deficit, the vestibular signals entering the CNS during head rotations are asymmetrical because of the excitatory domain of the response. Therefore, the patient receives a continuous asymmetrical vestibular signal that can internally construct and maintain a perceptual impairment. The fact that vision failed to attenuate or cancel maladaptive responses is a further finding that supports the persistence of the perceptual error over time.

A further result of this research is the idea that it may be possible to eliminate the maladaptation to asymmetric vestibular stimulation. Cancellation can be achieved by applying asymmetric rotation with fast and slow half-cycles in the direction opposite to that of the conditioning. In fact, this stimulation completely abolished the perceptual changes in response to slow and fast cycles, indicating that the motion misperception induced by asymmetrical vestibular signals can be cancelled out by opposite signals activating the same vestibular circuitry. Interestingly, the cancellation occurred not only when conditioning was performed in the dark, but also in the light. This might suggest that specific vestibular training using asymmetrical rotation with a fast stimulus toward the side of the lesion could be useful in correcting the perceptual imbalance in patients with unilateral vestibular deficit.

**Author Contributions:** Conceptualization, V.E.P., M.F. and G.R.; methodology, V.E.P., R.P., A.F. and F.M.B.; formal analysis, F.M.B.; investigation, C.O. and R.P.; data curation, R.P., C.O., A.F. and F.M.B.; writing—original draft preparation, V.E.P.; writing—review and editing, V.E.P., M.F. and G.R.; supervision, G.R. and V.E.P. All authors have read and agreed to the published version of the manuscript.

**Funding:** This research received no external funding.

**Institutional Review Board Statement:** The study was conducted in accordance with the Declaration of Helsinki and approved by the Institutional Review Board (or Ethics Committee) of CER Umbria (protocol code Nashville 30/6/21).

**Informed Consent Statement:** Informed consent was obtained from all subjects involved in the study.

**Data Availability Statement:** The data presented in this study are available upon request from the corresponding author.

**Conflicts of Interest:** The authors declare no conflict of interest.

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
