# Peer review of "Induction and Cancellation of Self-Motion Misperception by Asymmetric Rotation in the Light"

_audiolres, doi:10.3390/audiolres13020019_

Round 1

Reviewer 1 Report

General comments:

This study aimed to assess if an asymmetric sinusoidal whole-body paradigm can impact motion perception in healthy adults when delivered in the light and the dark. Overall, they were able to identify that changed in motion perception were not seen when assessed in the light, but this stimulus still altered symmetry of motion perception. There are several concerns that need to be addressed.  Firstly, “in the light” is broadly used throughout the manuscript. This implies that just light is driving this effect and not (congruent) visual input—using this nomenclature, this suggests that even performing this experiment with a headset that somehow provided illumination without broader visual input would induce the same results.  Also, what testing conditions that were performed was extremely unclear and made the results (and conclusions) very hard to follow.  As well, much of the methodology was just fully justified, and I was unclear why some of the procedures were being done. There are also several claims about potential applications in patient populations, but testing was only done in healthy controls, thus these cannot be fully substantiated.

Title “possible procedure for compensation after vestibular damage”, this is not fully supported since this was not tested in vestibular loss and was only shown in healthy controls.

Introduction:

The fact that a conditioning paradigm is being used is only addressed in sentence 51-52—this should be mentioned earlier.

“Line 56-57 “suggesting a procedure for restoring normal self-motion perception in vestibular deficit” is too much of a reach as this was only in healthy controls.

Methods:

Participants: how were the participants declared healthy? What symptoms/diseases were they screened for?

Section 2.2. “Self-motion perception condition” During the conditioning stimulus they also did the tracking?

125-134: The testing session descriptions may be better described in a table or diagram as this is extremely hard to follow.

Is there a reason the asymmetric rotation was always performed in the light/dark which matched the conditioning stimulus? If the goal was to look at the impact of vision, why was the testing after the conditioning only done in the dark? And not also the light?

Results:

Section 3.3: Why was the asymmetric rotation always performed in the same light condition as the conditioning stimulus?

Figure 5: Is there no data from the oppositely directed stimuli to show? Why is this only TPE and FPE?

Can there be a figure to show the comparison between the light/dark to the other conditioning stimuli without the oppositely directed stimuli?

Dicussion:

In general, this paradigm only stimulate the horizontal SCC, which should be acknowledged. Also, some discussions on the impact (or lack of impact) of velocity storage should be discussed. As this study was only conducted in healthy adults, potential applications to vestibular loss are not clearly known and conclusions should be discussed in this light more clearly.

Lines 270-274: How a unilateral deficit and this asymmetric paradigm are related is not clearly defined.

Reviewer 2 Report

Very interesting well written study. However, I have raised the following points:

1. The introduction can be improved. This is a complex topic and concepts and ideas are not presented succinctly and are overtly technical. For this paper to appeal to a wider audience this needs to be simplified.

2. The selection of 'healthy' needs to be defined. Even a minor aberration of central function can skew results. All subjects must have scored adequately on a cognitive score and screened for at least low frequency vestibular function.

3. The discussion section again lacks the simplicity and clarity needed.

4. I have annotated my comments in the attached file.

Round 2

Reviewer 1 Report

Overall, the implemented edits addressed many concerns and improved overall clarity.  A few minor concerns remain:

Lines 57-59: …” at the base of the maintained self-motion misperception”.  Please clarify what is meant by this sentence.

Lines 25-62: breaking this into two (or more) paragraphs may make this easier to parse.

Lines 75:  Were the “conventional tests” performed focused on the horizontal SCC? (e.g., calorics, vHIT, rotational testing…)

Lines 108-109: “Both acceleration and velocity values are well above the thresholds for vestibular activation.”

Please provide citation.

Lines 155-167: The different procedure are referred to as “sessions” in text in the methods, however, nothing else is ever referred to as “session” in text or figures throughout the rest of the paper. To minimize repetition and for clarity, consider incorporating this more broadly in text and Figures. Keeping track of which condition is being discussed can be confusing at times.

Section 3.1: Are the magnitude of these results in line with past results using the same paradigm? As session 1 was to confirm asymmetric rotation leads to perceptual adaptation, this is a good thing to note/cite.

Lines 282-284: “This effect may be the result of intermittent activation of a velocity storage mechanism in the perceptual circuitry of the horizontal semicircular canals, a mechanism like that observed in the oculomotor system, but much more durable.”

The last half of the sentence is unclear/takes some parsing. This is more durable for the VOR or perception?

Lines 304-308: This paragraph is almost verbatim from the Introduction. One of the sections would benefit from being re-worded.

Author Response

Answer to referee.

Overall, the implemented edits addressed many concerns and improved overall clarity.  A few minor concerns remain:

Lines 57-59: …” at the base of the maintained self-motion misperception”.  Please clarify what is meant by this sentence:.

This part of the sentence has been eliminated, since it was not necessary. We changed the sentence:

the induction and maintenance of a maladaptive mechanism implies that vision is not able to abolish the effect of the erroneous motion perception of vestibular origin”

Lines 25-62: breaking this into two (or more) paragraphs may make this easier to parse.

The introduction has been separated in more paragraphs.

Lines 75:  Were the “conventional tests” performed focused on the horizontal SCC? (e.g., calorics, vHIT, rotational testing…)

Most of the conventional tests are for horizontal SCC, but we include also the SVV. This is in the methods.

Lines 108-109: “Both acceleration and velocity values are well above the thresholds for vestibular activation.” Please provide citation.

The citations have been included

Lines 155-167: The different procedure are referred to as “sessions” in text in the methods, however, nothing else is ever referred to as “session” in text or figures throughout the rest of the paper. To minimize repetition and for clarity, consider incorporating this more broadly in text and Figures. Keeping track of which condition is being discussed can be confusing at times.

The different sessions have been indicated either in the results and in the captions

Section 3.1: Are the magnitude of these results in line with past results using the same paradigm? As session 1 was to confirm asymmetric rotation leads to perceptual adaptation, this is a good thing to note/cite.

We included a sentence for underline the consistency of the present results with the previous ones.

The amplitude of the TPE and FPE observed after repetitive asymmetric rotation was consistent with that observed in previous studies (1-3). “

Lines 282-284: “This effect may be the result of intermittent activation of a velocity storage mechanism in the perceptual circuitry of the horizontal semicircular canals, a mechanism like that observed in the oculomotor system, but much more durable.”

The last half of the sentence is unclear/takes some parsing. This is more durable for the VOR or perception?

Sentences have been changed.  “However, the perceptual effect of the intermittent activation is much more durable. The perception change persists for more than 30’ minutes and displaces the internal representation of the body position in space, while the velocity storage observed in the ocular responses is remarkable shorter. This highlights that the central elaboration of the vestibular signals directed to the motion perception is different from that directed to the oculomotor system”

Lines 304-308: This paragraph is almost verbatim from the Introduction. One of the sections would benefit from being re-worded.

The sentences have been changed. “The higher sensitivity of the vestibular system for fast movements in combination with the visual and proprioceptive systems that are sensitive for slow movements [1,29-30], is functionally useful to enhance the self-motion perception.”
